



# Local turbulence parameterization improves the Jensen wake model and its implementation for power optimization of an operating wind farm

Thomas Duc[1], Olivier Coupiac[1], Nicolas Girard[1], Gregor Giebel[2], and Tuhfe Göçmen[2]

[1]ENGIE Green France, 59 rue Denuzière, 69002 Lyon, France
[2]DTU Wind Energy, Risø Campus, Frederiksborgvej 399, DK 4000 Roskilde, Denmark

*Correspondence to:* Thomas Duc (thomas.duc@engie.com)

**Abstract.** In this paper, a new calculation procedure to improve the accuracy of the Jensen wake model for operating wind farms is proposed. In this procedure the wake decay constant is updated locally at each wind turbine based on the turbulence intensity measurement provided by the nacelle anemometer. This procedure was tested against experimental data at onshore wind farm La Sole du Moulin Vieux (SMV) in France and the offshore wind farm Horns Rev-I in Denmark. Results indicate

that the wake deficit at each wind turbine is described more accurately than when using the original model, reducing the error from 15 - 20% to approximately 5%. Furthermore, this new model properly calibrated for the SMV wind farm is then used for coordinated control purposes. Assuming an axial induction control strategy, and following a model predictive approach, new power settings leading to an increased overall power production of the farm are derived. Power gains found are in the order of 2.5% for a two wind turbine case with close spacing and 1 to 1.5% for a row of five wind turbines with a larger spacing. Finally,

the uncertainty of the updated Jensen model is quantified considering the model inputs. When checked against the predicted power gain, the uncertainty of the model estimations is seen to be excessive, reaching approximately 4%, which indicates the difficulty of field observations for such a gain. Nevertheless, the optimized settings are to be implemented during a field test campaign at SMV wind farm in scope of the national project SMARTEOLE.

## 1 Introduction

Wind turbines are aggregated together in wind farms to take advantage of economies of scale and reduce overall costs (Pao and Johnson, 2009). However this creates wake interactions between the turbines which are responsible for an increase in mechanical loads and a decrease in power production. It is generally not possible to avoid completely these interactions due to constraints imposed on the development of wind farms and moreover in some cases wake effects are still persistent at significant distances downstream (Sanderse, 2009).

To reduce these effects and improve wind farm efficiency and sustainability, wind farm coordinated control strategies are currently investigated. Contrary to the state-of-the-art control, in which all turbines maximize their own power production, coordinated control aims at controlling turbines at a wind farm scale to optimize its overall output. Two different strategies are





mainly considered to achieve this goal: either the upwind turbines are curtailed to leave more kinetic energy downstream or they are yawed to deflect the wake away from the downwind turbines.

Results from simulations show that small gains in power production are indeed possible (Bossanyi and Jorge, 2016; Gebraad et al., 2017), however they also underline their high variability with incoming wind conditions (Knudsen et al., 2015). It

is therefore not known to what extent these gains can be reproduced in an operating wind farm where wind conditions are fluctuating constantly and significantly. Very few full scale field tests have been realized to investigate this question. The concepts of "Heat and Flux" (Machielse et al., 2007) and "Controlling Wind" (Wagenaar et al., 2012) were studied some years ago at the Energy Center of the Netherlands (ECN) and more recently the National Renewable Energy Laboratory (NREL) provided in Fleming et al. (2017) a field test of their "yaw-based wake steering" method in an Envision offshore wind farm.

They tend to confirm that gains can be achieved in practice, however in all cases uncertainties remain high and it is therefore difficult to give a definite conclusion.

Other full scale field tests are currently being held in France in the scope of the French national project SMARTEOLE. They are organized in an operating wind farm owned by ENGIE Green, La Sole du Moulin Vieux (SMV), in which different curtailment and yaw offset strategies are studied. The main objective of these tests is to investigate the relevance of these

strategies on wind turbine power production and loads and determine whether they could prove beneficial when applied on commercial wind farms. A first experiment campaign was realized between December 2015 and April 2016 and was dedicated to axial induction control strategy. An intentionally high level of curtailment was applied on a wind turbine of the farm so that changes in its emitted wake could be observed. Even though no increase in combined power production was expected, a first analysis of wind turbine power production in the farm showed that part of the lost power at the upstream turbine could be

retrieved downstream (Ahmad et al., 2017).

The goal of this paper is to use the knowledge gained during the first campaign to provide new optimized control commands that could be implemented in second field campaign, and hopefully lead to an increase in combined power production of the farm. As SMV is a commercial wind farm, these commands must be easily applicable without modifying the wind turbine control system, thus a model predictive approach is followed to determine the best settings as a function of wind speed and

direction. To limit computational costs and the complexity of the optimization process, fast and simple models are considered. Hence simplified engineering models are to be applied and the main issue when following this kind of approach is making sure that they are capturing accurately the wake deficit at each wind turbine. Consequently the data recorded during the first field test campaign is analyzed further and used to propose a modification of the widely used Jensen model. In this new method the wake decay constant is expressed at each wind turbine based on the local measurement of turbulence intensity. The resulting

wake deficit appears to be more consistent with observed data at SMV wind farm than when using the original model, and this calculation procedure is also validated considering experimental data from the Horns Rev-I offshore wind farm.

The rest of this paper is organized as follows. In Sect. 2, the experimental setup used during the first field test campaign is shortly detailed. Section 3 describes the principle of the modified Jensen model, and its performance compared to the original model is assessed in Sect. 4. This modified model is then used in Sect. 5 alongside with a simple $c_T$ estimation procedure to

predict in two study cases at SMV wind farm the optimized settings leading to an increase in overall wind farm production.



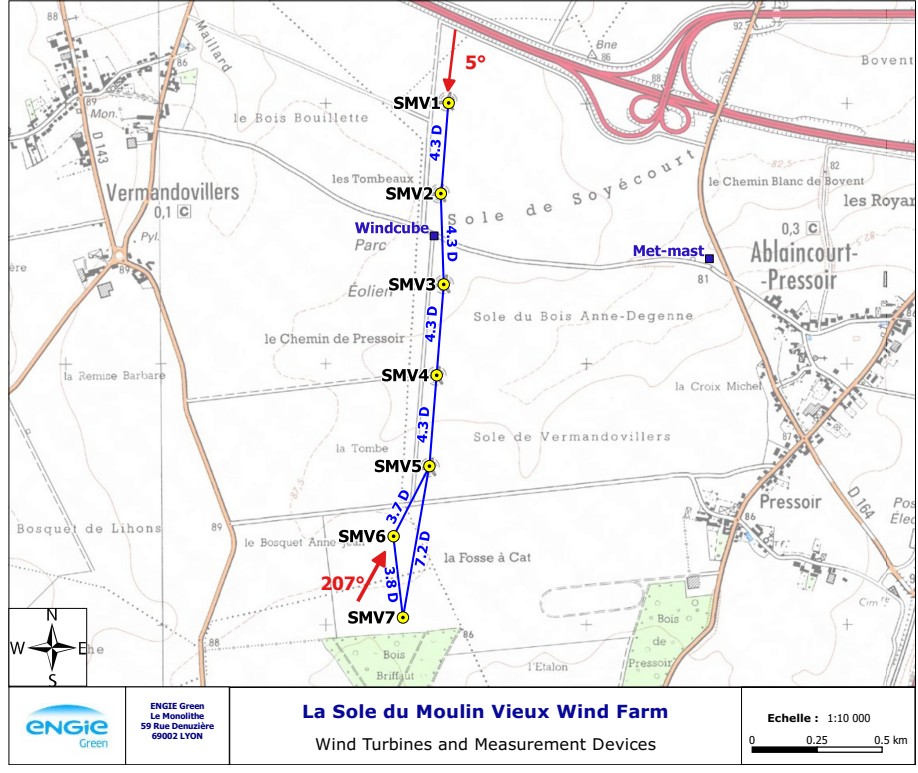

**Figure 1.** Layout of SMV wind farm and location of wind measurement devices. Inter distances between the wind turbines are expressed in rotor diameters, while red arrows indicate the main wind direction used in this paper.

Uncertainty of the model estimation is also quantified in this section. Finally, Sect. 6 provides a summary of the paper and conclusions.

## 2   Experimental setup

La Sole du Moulin Vieux is a commercial wind farm owned by ENGIE Green and located at Ablaincourt-Pressoir in the region
5   Hauts-de-France, approximately midway between Paris and Lille. Figure 1 shows the layout of the farm with the inter-distances between the turbines and main direction angles used in this paper. It consists of 7 Senvion REpower MM82 2050 kW wind turbines 80 m hub height, that were commissioned in two steps: the first five turbines (SMV1 to SMV5) were put in service in 2009 while the two last ones (SMV6 and SMV7) were installed four years later in 2013. The site is not complex, with a very flat terrain composed mainly of grasslands, with the exception of a small wood located south of the farm.
10      It can be seen that wind turbines are more or less aligned on a North-South axis while prevailing wind direction is South-West, as seen on the long-term wind rose of Fig. 2(a). This particular wind farm was chosen for the field test campaigns of the SMARTEOLE project because of the proximity with ENGIE Green maintenance center (located 5 km away from the farm) and

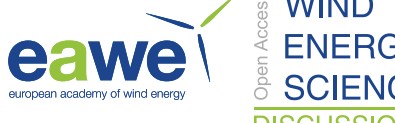



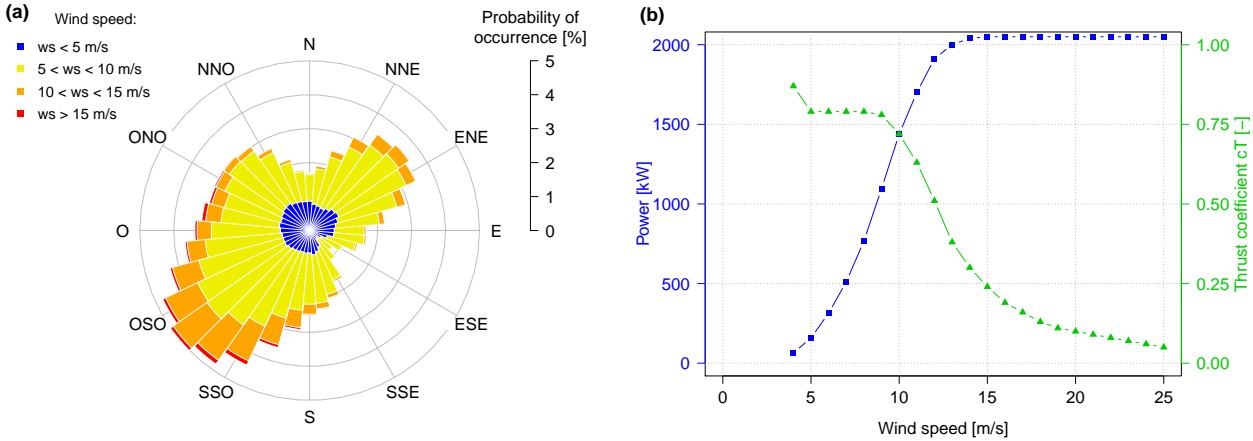

**Figure 2.** Long term wind rose observed at the site of SMV wind farm (a) and Senvion MM82 guaranteed power and thrust coefficient ($c_T$) curves (b).

the wake event SMV6 - SMV5. Indeed, due to development constraints, these two turbines were installed very close from each other (only 305 m, i.e. 3.7 D) and aligned with prevailing wind directions. In this paper SCADA data from the seven turbines is analyzed along with data from a 80 m met-mast and a ground-based lidar (windcube V1). The location of these sensors is indicated on Fig. 1.

## 3 Modification of the Jensen model

### 3.1 Original model

The Jensen model as it was originally developed by Jensen (1983) and Katic et al. (1986) is introduced briefly here. In this model the wake grows linearly at a rate driven by a coefficient $k_w$ called wake decay constant (WDC) or wake expansion coefficient. The wind speed deficit $\delta_w$ in the wake is assumed to be uniform, axis-symmetric and depends only on the downstream distance $x$ and the upstream wind turbine thrust coefficient $c_T$. It can be computed using mass conservation and can be expressed as:

$$\frac{U_w}{U_0} = 1 - \delta_w = 1 - \frac{1 - \sqrt{1 - c_T}}{(1 + k_w x/R)^2},\tag{1}$$

where $U_0$ is the incoming wind speed at the upstream wind turbine, $U_w$ the velocity in the wake and $R$ the radius of the upstream rotor.

When summing the wakes from two or more upwind rotors, wind speed deficits are aggregated via quadratic sum. Therefore the wake deficit $\delta_n$ at the n-th wind turbine of a row is simply given by:

$$\delta_n = \sqrt{\sum_{i=1}^{n-1} \delta_i^2},\tag{2}$$





where $\delta_i$ is the wind speed deficit due to wind turbine i.

As indicated earlier, the Jensen model is probably the most widely-used wake model for wind energy engineering applications, and that is mainly due to its simplicity and robustness. In particular, its very low computational cost makes it very suitable for optimization purposes as a high number of simulations can be run in a very short time. However, although this model gives fairly good estimation of the averaged power deficit in a wind farm (Göçmen et al., 2016), some studies underline its inaccuracy when it comes to looking at the individual power production of the wind turbines (Barthelmie et al., 2009; Gaumond et al., 2014). This is a crucial issue for power optimization using coordinated control, given that the production at the downstream turbines must be predicted as accurately as possible in order to choose the optimal settings of the upstream turbine.

There is therefore a need for improvement to be able to use this model for such purposes. Over the past years some new models have been derived from the main equation of the Jensen model. They aim at offering a better representation of the individual wake deficits by introducing new parameters and equations, while keeping a low computational cost. For example, in Bastankhah and Porté-Agel (2014) the equation of momentum conservation is included to the model and a Gaussian distribution is assumed for the velocity deficit profile. The multizone model developed by Gebraad et al. (2014) is also Jensen-based and considers three different areas in the wake, each with its own wake decay constant. Description of wind turbine wakes is indeed very much improved with these models, however they can be relatively difficult to calibrate as they consider up to ten parameters that need to be tuned properly (Annoni et al., 2018).

In this paper a very simple tuning of the original Jensen model is proposed based on the measure of local turbulence intensity (TI). The idea is to keep the simplicity of calibration and robustness of the model while improving its accuracy. As it will be discussed in the next section, and shown later in Sect. 4, taking turbulence intensity into account when tuning the model already improves significantly the performance of the model and offers a fairly good representation of the velocity deficit along a row of turbines, both onshore and offshore.

## 3.2 Tuning of the model

As can be seen in Eq. 1, there is only one parameter to be tuned in the Jensen model: the wake decay constant. This empirical constant is supposed to vary from one wind farm to another but generally the two recommended values of 0.075 and 0.05 are used for onshore and offshore wind farms, respectively (Mortensen et al., 2011). In some studies it is also expressed more specifically as a function of the particular conditions at the wind farm, using for example the roughness length and the atmospheric stability (Peña and Rathmann, 2014) or the ambient turbulence intensity (Peña et al., 2015; Thorgersen et al., 2011). Recent studies, based on wind tunnel, Large Eddy Simulations (LES) and full scale turbine data, clearly identifies TI as one of the most influencing parameter on the wake growth and magnitude of the wake deficits (Bastankhah and Porté-Agel, 2014; Mittelmeier et al., 2017; Annoni et al., 2018).

It is known that TI varies significantly inside a wind farm, as the wake-added TI from upstream wind turbines is added to the ambient TI (Crespo et al., 1999; Vermeer et al., 2003; Göçmen and Giebel, 2016). Keeping the same wake decay constant for all wind turbines in the farm can therefore lead to errors in prediction of individual power production. The wake deficit is under-estimated at the first few downstream turbines and over-estimated further down the row (Gaumond et al., 2014) or vice





versa (Göçmen et al., 2016). Consequently, it appears more accurate to assign a new wake decay constant for each wind turbine that would be directly linked to the local value of TI rather than considering an averaged value for the complete wind farm.

This strategy was followed in Niayifar and Porté-Agel (2016) and an expression between the wake decay constant and the local TI was proposed on the basis of LES data:

$$k_w = 0.3837 \, (TI)_{mod} + 0.003678, \tag{3}$$

where $(TI)_{mod}$ is the modeled local TI obtained through the combination of ambient and wake added TI, the latter being estimated thanks to the empirical equation developed by Crespo et al. (1996). As in this present paper SCADA data is available, it was rather decided to use the direct measurement of turbulence intensity provided at each wind turbine rather than relying on such a generic expression. In the next section are presented several methods that can be used to estimate TI from SCADA. A direct proportionality is considered between $k_w$ and the measured TI $(TI)_{meas}$. This was done in order to keep the same simplicity as in the original Jensen model, i.e. to have only one parameter to calibrate, the constant $c$:

$$k_w = c \, (TI)_{meas}. \tag{4}$$

It should be noted that the tuning of the wake decay constant is based on TI only. Although TI is probably the most critical parameter, some studies underline the link between the variation of $c_T$ and $k_w$ (Bastankhah and Porté-Agel, 2014; Annoni et al., 2016). Accordingly in Sect. 4 the tuning of the Jensen model is evaluated in constant $c_T$ region, see Fig. 2, in order to isolate the local turbulence effects on the wake expansion. However, this is no longer true in Sect 5 where the model is used in an optimization process involving axial-induction control, whose purpose is precisely to change the $c_T$ of upstream turbines.

### 3.3 Estimating TI from SCADA

There are several ways to assess the incoming wind speed from ordinary SCADA signals, all with their advantages and drawbacks. These different methods are presented in this section and they can all be used to derive a local estimation of TI which is a required input for the wake model presented above. Alternative methods using sensors that are usually not installed on wind turbines (e.g. lidars or spinner anemometers) are not developed here but could also fulfill the same purpose.

The first and most obvious way to obtain a wind speed measurement from SCADA is to consider the nacelle wind speed signal (NWS) emitted by the nacelle anemometers installed on the wind turbine. However, these sensors are located behind the rotor and are therefore exposed to a highly distorted flow (Zahle and Sørensen, 2011). They cannot be relied on to provide an accurate and instantaneous wind speed measurement, but when it comes to considering local TI they might be good enough as only 10-min average and standard deviation values will be involved.

Another wind speed measurement can easily be obtained from the active power production signal and the guaranteed power curve of the wind turbine (or a measured power curve). This new signal, labeled here as power curve wind speed (PWS), is generally more reliable than the NWS since the sensor is the wind turbine itself. Also, due to rotor inertia and the fact that the wind speed derived using this method is averaged over the whole rotor area, small fluctuations of the wind flow will be filtered out resulting in a much smoother signal. The main issue regarding this method is its limited applicability: it cannot be used above rated wind speed or during down-regulation, therefore unsuitable for wind farm coordinated control purposes.





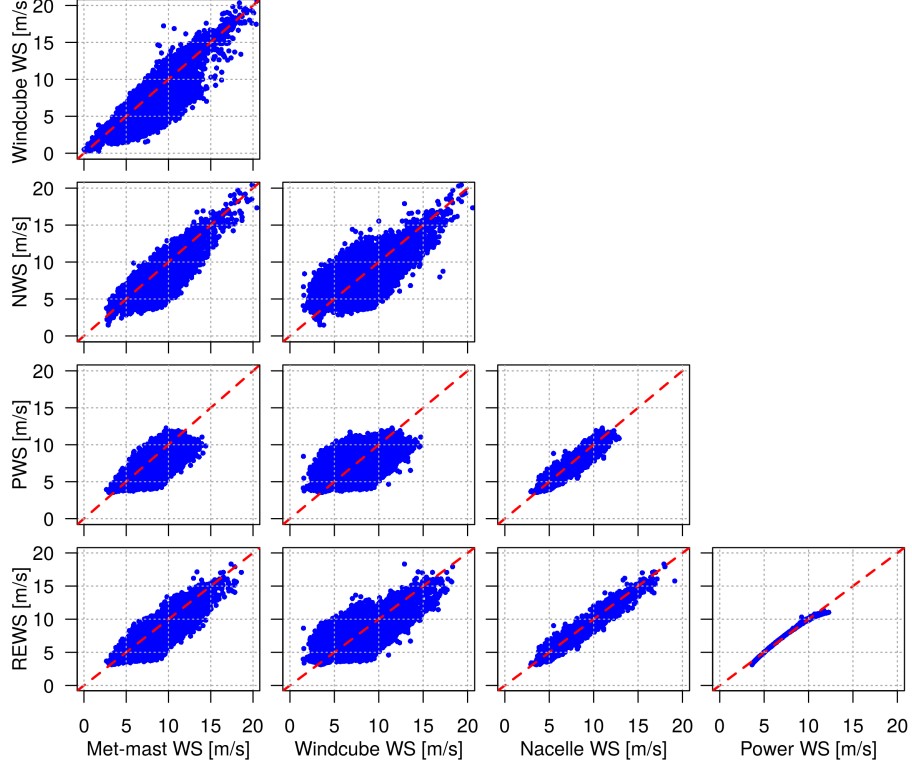

**Figure 3.** Comparison of wind speeds measurements at SMV5 (for Nacelle Wind Speed, Power Wind Speed and Rotor Effective Wind Speed) and at external sensors (met-mast and windcube). No sector filtering was applied to the data, explaining the scatter when comparing sensors at different locations.

In order to solve this problem a third way was developed in the scope of the PossPOW project to calculate the rotor effective wind speed (REWS) of a wind turbine in these particular situations by Göçmen et al. (2014). In this method the REWS is calculated from three SCADA signals (active power, rotor speed and pitch angle) and a $c_P$ model. It must be ensured that the chosen $c_P$ model is fitting as best as possible to the real $c_P$ curve. Alternatively a $c_P$ look-up table can also be used, when available. In this paper the $c_P$ model and the value of its parameters are the same as the one presented in Göçmen (2016), as they proved to offer a good performance for wind turbines in the same range (rated power of 2 MW with a diameter of about 80 m) as the ones studied here.

Four months of second-wise SCADA data were processed (1st December 2015 - 31st March 2016) for the wind turbine SMV5 to compute 10-minutes average wind speed and turbulence intensity using these three different methods. Figure 3 compares these wind speed values with measurements from reference sensors (met-mast and windcube). No sector filtering was applied to the data, consequently a significant scatter can be found when comparing two sensors at different locations due to wake effects. It can be observed that these results are very similar to the ones that were obtained at the Lillgrund offshore wind farm and were presented in Göçmen and Giebel (2016), showing a very nice correlation between the PWS and the REWS, and more




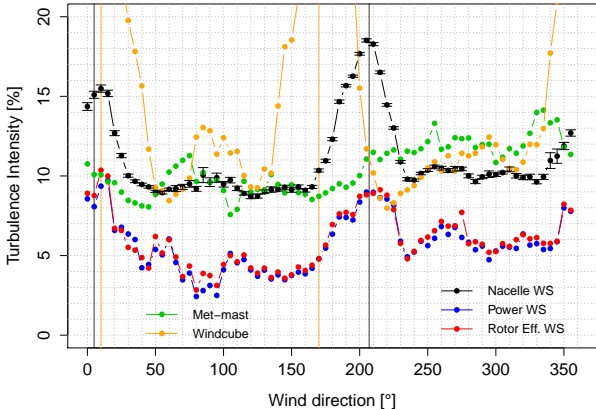

**Figure 4.** Comparison of TI measurements at SMV5 (for Nacelle Wind Speed, Power Wind Speed and Rotor Effective Wind Speed) and at external sensors (Met-mast and windcube), represented against wind direction. Error bars shows the 68% normalized confidence interval (represented for NWS but a same order of magnitude can be expected for other sensors in the same direction bin). Vertical lines indicate position of wakes for SMV5 sensors and windcube.

scatter when considering the NWS. This is because the wind speeds calculated through the REWS or PWS methods contain a geometrical averaging of the wind flow over the whole surface of the rotor which smooths out wind speed fluctuations (Göçmen and Giebel, 2016). On the other hand NWS and met-mast are point-wise measurements and therefore are affected by every single variation of the wind speed.

5     In Fig. 4 the turbulence intensity calculated from all these signals is represented against wind direction (5° bins). It can be seen that outside any wake events, the TI obtained through the NWS signal is of same order of magnitude than the one measured by external sensors. On the contrary, TI obtained with either the PWS or the REWS signals is approximately twice as low. As previously mentioned, this is explained by the geometrical averaging provided by these two methods.

    All wake events are clearly captured by any of the TI signals. At a particular location of a wake, TI measured is about twice
10  as high compared to the free-stream conditions, indicating that wake added TI is significant and can be measured reliably with a SCADA signal. Consequently any of these TI signals could be used as input for the modified wake model, it is simply needed to adjust accordingly the value of the constant $c$ in Eq. 4. However, for the rest of this paper, only the NWS TI could be considered in practice given that the PWS signal cannot be used for down-regulation purposes, while the REWS method requires acquisition and processing of second-wise data, which were not available for some of the turbines in the wind farm.

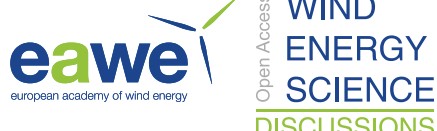

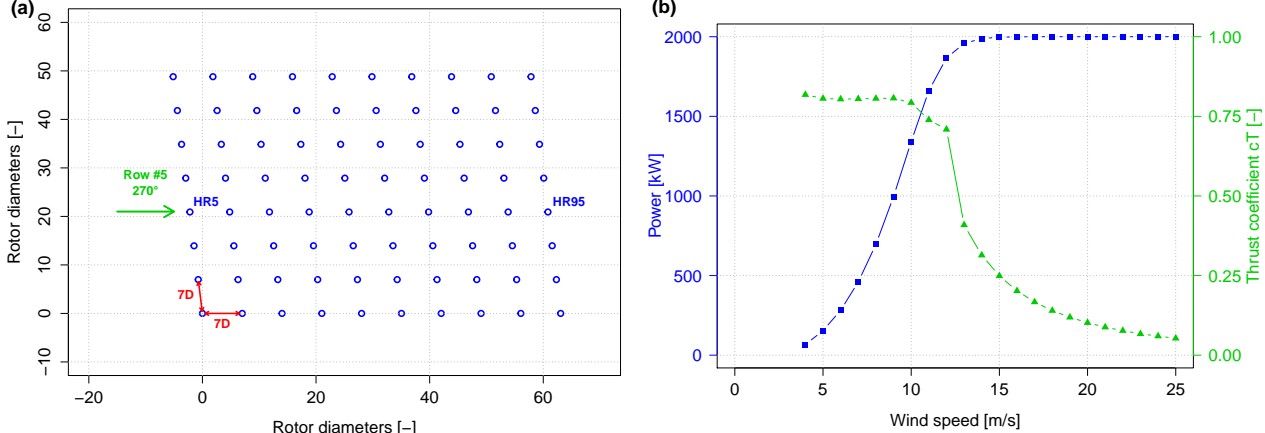

**Figure 5.** Layout of Horns Rev-I offshore wind farm with turbine spacing and wind direction angle used in ths study (a) and Vestas V80-2MW guaranteed power and thrust coefficient ($c_T$) curves (b).

## 4   Validation of the tuning strategy

### 4.1   Data filtering and processing

The performance of the modified Jensen model is now assessed in this section and compared to the results obtained for the original model. Production data from two wind farms are considered: the onshore SMV wind farm and the offshore wind farm of Horns Rev-I (layout of the farm and power and $c_T$ curves for the Vestas V80-2MW wind turbines are shown on Fig 5). In both case, the normalized power production along a row of turbines is analyzed. The data is filtered to keep only the 10-minutes periods when all wind turbines are in operation and the incoming wind direction is within a $\pm 5°$ interval around the main orientation of the row. Another filtering is done based on the power production of the most upstream turbine in order to consider only the 10-min periods when wind turbines are all in the constant $c_T$ region (the Region II of the power curve).

For each valid 10-min period, the power production deficit at each wind turbine is simulated for both the original and the tuned models. When considering the original Jensen model, the value of $k_w$ is calibrated using the first wake event of the row. In the case of the tuned wake decay constant, the value of the constant $c$ in Eq. 4 is adjusted roughly to limit the individual error of the model along the row. The value used for $(TI)_{meas}$ is the 10-min TI measurement from the NWS signal. The normalized simulated deficits are then averaged over all valid 10-min periods and compared with the normalized measured deficits. The averaged value of the NWS TI signal at each wind turbine is also drawn to show the variation of the measured TI along the row of turbine. Error bars indicate the 68% normalized confidence interval.

At SMV wind farm, the power production deficit is studied along the row SMV1 to SMV5. The wind direction sector considered is $5 \pm 5°$, and the valid power range for the most upwind wind turbine (SMV1) is fixed to 600 - 1100 kW to fulfill the constant $c_T$ condition. The filtered data-set finally consists of 156 valid 10-min periods recorded between 15th June 2015 and 1st August 2016 (due to small occurrence of northerly winds, it was needed to consider a longer period than the actual field

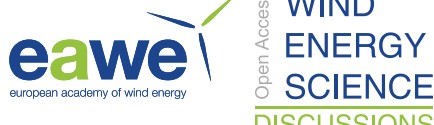



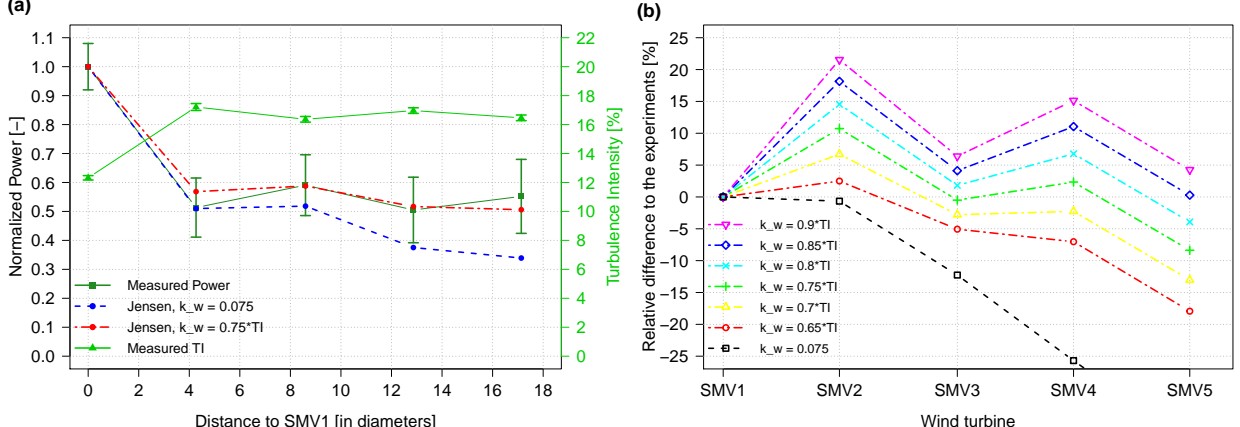

**Figure 6.** Comparison of the model performance at SMV wind farm. Variation of experimental and modeled normalized power and turbulence intensity along the row (a) and variation of the error of the model at each wind turbine for various values of wake decay constants $k_w$ (b). Error bars indicates the 68% normalized confidence interval.

tests to gather enough valid data). The wake decay constant for the original Jensen model was kept as 0.075, the traditional value for onshore wind farm as it proved to show a good performance for the first wake event of the row SMV1 - SMV2.

At Horns Rev-I wind farm, the power production deficit is studied along the row number 5, from HR5 to HR95 (see layout of the farm on Fig. 5(a)). The wind direction sector considered is $270\pm5°$, and the valid power range for the most upwind wind turbine (HR5) is fixed to 0 - 1200 kW to fulfill the constant $c_T$ condition. The filtered data-set finally consists of 270 10-minutes periods recorded between 16th February 2005 and 25th January 2006. The value for the wake decay constant of the original Jensen was fixed to 0.09, which is much bigger than the value of 0.05 traditionally used for offshore wind farms but much more consistent with the measured deficits. It was already found on other studies (e.g. Niayifar and Porté-Agel (2016)) that using a wake decay constant of 0.05 was clearly overestimating the power deficit for the wind farm of Horns Rev-I, as it gives narrower wake growth within the wind farm.

### 4.2 Results

The evaluation of the original and the calibrated Jensen model on SMV and Hons Rev wind farms are presented on Fig. 6 and 7, respectively. In both cases the normalized deficit is shown as a function of distance to the most upstream turbine, as well as the difference between the modeled and the observed deficit at each wind turbine for different values of wake decay coefficient. It can be seen that the graphs for the two wind farms have a very similar behavior with the tuned Jensen model performing better than the original model. Except for the first wind turbine, the wake deficit calculated with the original model is always overestimated and the error is getting more significant towards downstream: from approximately 10% at the third turbine of the row, it goes around 15 to 20% further downstream. On the contrary, with the tuned Jensen model the deficit is captured more accurately, especially at the wind turbines located at the middle of the row for which the error is kept between $\pm5\%$. These

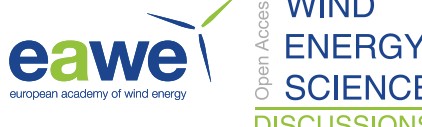


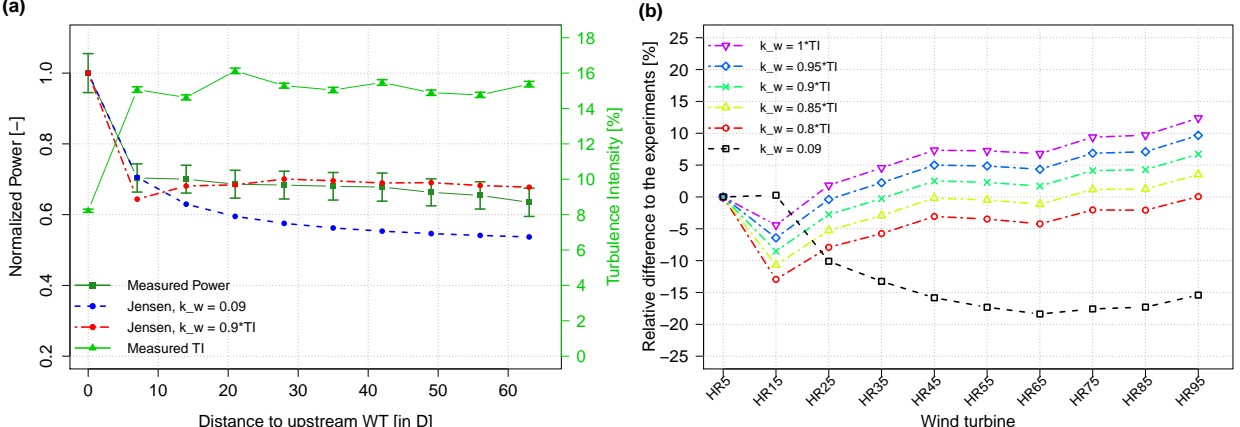

**Figure 7.** Comparison of the model performance at Horns Rev-I wind farm. Variation of experimental and modeled normalized power and turbulence intensity along the row (a) and variation of the error of the model at each wind turbine for various values of wake decay constants $k_w$ (b). Error bars indicates the 68% normalized confidence interval.

changes are consistent with the augmentation of turbulence intensity from the second wind turbine in the row. Increased TI provides a better mixing between the disturbed flow in the wake and the undisturbed free flow, which allows a earlier recovery, both in terms of time and space. This particularity is taken into account in the tuned Jensen model since the WDC is increased linearly with TI, while with the original Jensen keeping the same WDC all along the row leads to an overestimation of the

5 deficit.

When analyzing the impact of the choice of the constant $c$ linking TI with the $k_w$ in the tuned Jensen model, two observations can be made. First, it can be seen that the optimal $c$ obtained for each wind farm is different, 0.75 for SMV wind farm and 0.9 in the case of Horns Rev-I. This shows the sensitivity of the model to the local conditions and tends to indicate that a small calibration of the constant will still be needed for each wind farm to make sure that the wake deficit is correctly modeled. This

site-dependency of the Jensen model was already present in its original form; indeed in this example it can be seen that the best WDC obtained for the offshore wind farm (0.09) is much higher than the recommended values (0.04 - 0.05) and than the one of the offshore wind farm (0.075).

The second observation is that the modeled wake deficit is varying regularly when $c$ changes. This ensures that the calibration is as easy and robust as before, since it is simply needed to tune the value of $c$ until the wake deficit is described as best

as possible. Contrary to the original model, the accuracy is improved as taking into account the local TI allows a better representation of the individual wake deficit at each wind turbine.

Consequently, it can be concluded that this tuned Jensen model is providing an improvement compared with the original model, while keeping the simplicity of calibration and robustness of the original model. It can thus be used to define control instructions, as developed in the next section.



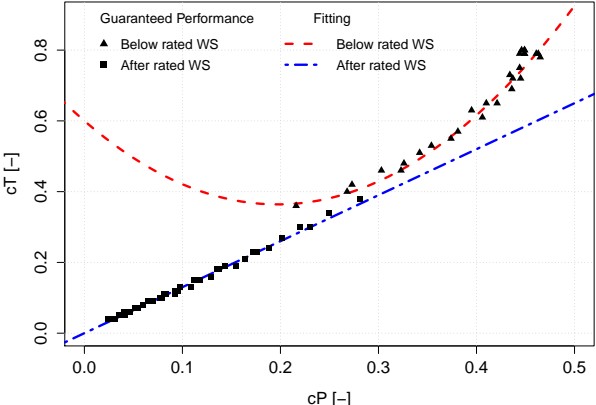

**Figure 8.** Correlation between $c_P$ and $c_T$ for a Senvion MM82 wind turbine, deduced from the analysis of guaranteed power curve modes for sound management.

## 5  Optimization of wind farm power production

After having validated the performance of the tuned Jensen model, it is used in an optimization process to find the wind turbine settings maximizing their power performance. Only the axial induction strategy is developed here, due to its ease of application on a commercial wind turbine. Indeed, it simply requires to trigger a pre-implemented down-regulated power curve
as a function of incoming wind conditions without modifying the control settings of the turbine. In practice only wind speed and direction can be used as input for triggering curtailment mode, therefore power production is optimized for various wind speed and direction bins.

The hypotheses used during the optimization process are first presented, and then two study cases at SMV wind farm are analyzed. Finally, a study of the uncertainty of the model outputs is realized based on the data from Horns-Rev I, and the values
obtained are compared with the predicted gains.

### 5.1  $c_T$ estimation procedure

The principle of wind farm power optimization using an axial induction control strategy is to curtail the upstream wind turbines to gain more energy on downstream wind turbines. It is hoped that the decrease in the upstream $c_T$ will be high enough to reduce sufficiently the wake deficit so that the increase in production downstream can compensate for the upstream $c_P$ diminution.
Consequently, it is a crucial issue to assess as accurately as possible how both $c_P$ and $c_T$ are being affected by the upstream down-regulation, in order to correctly estimate the overall production for various curtailment modes.

It is sometimes possible to use look-up tables to link $c_P$ and $c_T$ with operational parameters of the wind turbine, such as rotor speed and pitch angle. However for this work no look-up tables were available and therefore another method had to be developed. A workaround was finally found by the analysis of MM82 guaranteed curtailed power curves used for noise





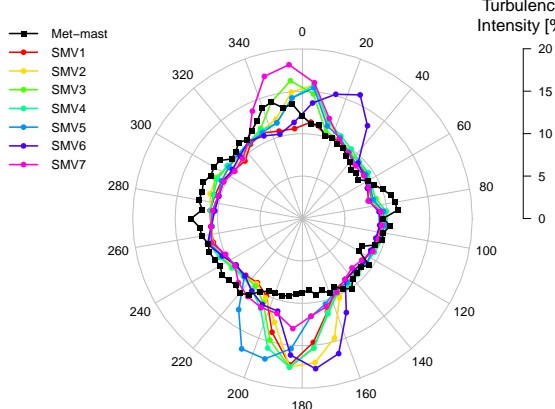

**Figure 9.** TI distribution at each wind turbine used during the optimization process. This rose is obtained for a wind speed of 8 m/s only, but similar roses were calculated for each wind speed bin. Ambient TI measured at the met-mast is also plotted for comparison.

emission reductions. Indeed, when representing $c_T$ against $c_P$, as in Fig. 8, two different behaviors are being observed: one below rated wind speed (parabolic relationship) and another after rated wind speed (linear dependency).

Thus an empirical relationship could be derived from this analysis, by fitting a second order polynomial to the first set of data points, and a first order polynomial for the second one. The final relationship between $c_P$ and $c_T$ used for the optimization

process and valid for an MM82 wind turbine is represented on Eq. 5 below.

$$
\begin{cases}
c_T = 6.1\,c_P^2 - 2.4\,c_P + 0.6 & \text{if } 6 < V < 12.5 \text{ m/s and } c_P > 0.2 \\
c_T = 1.3\,c_P & \text{if } V > 12.5 \text{ m/s}
\end{cases}
\tag{5}
$$

### 5.2 Turbulence intensity distribution

As developed in Sect. 3, knowing the turbulence intensity is of primary interest to assess properly the wake deficit. In Sect. 4, the TI was calculated in 10-min time scale resolution to compare the performance of the tuned wake model with the original one.

Here, due to practical constraints, it is not possible to consider the real-time TI as input parameter for triggering a curtailment mode. Instead, it was decided to express the local TI as a function of wind speed and direction by calculating a TI distribution in the farm. Consequently a different WDC is chosen for each wind turbine and each wind speed and direction bin, so that local TI still has some influence in the optimization process.

This TI distribution was obtained by averaging the NWS signal of all wind turbines in the farm in 10° direction bins and

1 m/s wind speed bins. The dependency of incoming TI for each wind turbine is represented for a wind speed of 8 m/s on a turbulence intensity rose in Fig. 9, alongside with TI measured by the met-mast that can be used for comparison.

It can be seen that the TI measured by the wind turbines outside any wake events is around 9 - 10%, which is consistent with the met-mast measurements. Some particular terrain effects can nonetheless be observed. Between 160° and 220°, the TI measured by SMV7 is increasing up to 12 to 13%: this was attributed to the presence of a wood south of the farm and very close





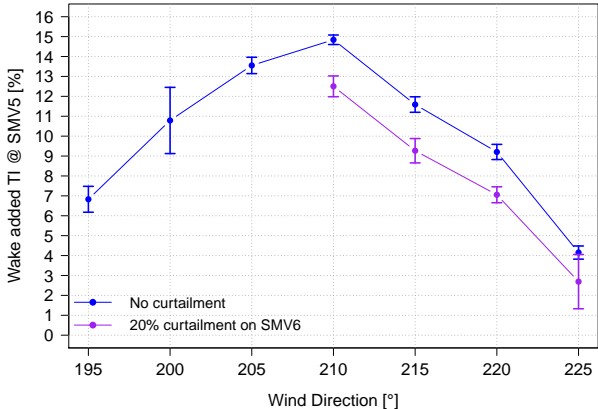

**Figure 10.** Variation of wake added TI as a function of wind direction, in normal operation and during SMV6 curtailment. Error bars indicate the 68% normalized confidence interval.

to SMV7 (please refer to the map of the wind farm on Fig.1). Likewise, an increase of 1 to 2% in the SMV1 TI is observed for wind direction between 340° and 20°. The reason for this increase was related the presence of the motorway at the north of the farm; indeed the met-mast curve shows also a similar increase in the sector [320°; 0°], corresponding to the direction of the motorway as seen by the met-mast location.

**5.3   Reduction of wake added TI**

In order to describe as accurately as possible the variation of TI in the farm when optimizing the power production, the influence of the upstream curtailment on downstream wake added TI must be taken into account. Indeed, as the upstream wind turbine is down-regulated, the wake added TI emitted by this turbine is reduced. It is expected that this decrease will be more and more significant as the upwind curtailment increases. Therefore the TI distribution presented in the previous section, and calculated

for normal operation conditions, is no longer valid: it must reduced accordingly with the percentage of down-regulation applied on the upwind turbine. This is important especially when considering a row of three turbines (or more), since the deficit at the third turbine is calculated based on TI at the second turbine, which is itself dependent on the first turbine curtailment.

To study the impact of upstream down-regulation on downstream wake-added TI, data from the first field test campaign is considered. Between December 2015 and April 2016, SMV6 was occasionally curtailed of approximately 20% for south-

western winds (in the direction of alignment SMV6-SMV5). Analyzing TI data provided by the nacelle anemometers, the wake added TI, $TI_{wa}$, can be computed with the following equation:

$$TI_{wa} = \sqrt{TI_{tot}^2 - TI_{amb}^2}, \tag{6}$$

where $TI_{tot}$ is the total TI in the wake measured at SMV5, and $TI_{amb}$ the ambient TI measured at SMV6.



**Table 1.** Comparison of measured wake added TI at SMV5 wind turbine between no curtailment and 20% curtailment on SMV6, with their 68% normalized confidence interval.

| Wind direction | No curtailment | | | 20% curtailment on SMV6 | | | Difference |
|---|---|---|---|---|---|---|---|
| [°] | $TI_{amb}$ [%] | $TI_{tot}$ [%] | $TI_{wa}$ [%] | $TI_{amb}$ [%] | $TI_{tot}$ [%] | $TI_{wa}$ [%] | $\Delta TI_{wa}$ [%] |
| 195 | $15.31 \pm 0.54$ | $16.76 \pm 0.37$ | $6.83 \pm 0.65$ | N/A | N/A | N/A | N/A |
| 200 | $13.20 \pm 1.43$ | $17.05 \pm 0.85$ | $10.79 \pm 1.66$ | N/A | N/A | N/A | N/A |
| 205 | $12.20 \pm 0.30$ | $18.24 \pm 0.28$ | $13.55 \pm 0.41$ | N/A | N/A | N/A | N/A |
| 210 | $11.80 \pm 0.16$ | $18.96 \pm 0.18$ | $14.84 \pm 0.24$ | $12.73 \pm 0.28$ | $17.84 \pm 0.44$ | $12.50 \pm 0.52$ | -2.34 |
| 215 | $11.90 \pm 0.26$ | $16.61 \pm 0.29$ | $11.59 \pm 0.39$ | $12.99 \pm 0.30$ | $15.96 \pm 0.53$ | $9.27 \pm 0.61$ | -2.32 |
| 220 | $11.51 \pm 0.23$ | $14.74 \pm 0.30$ | $9.21 \pm 0.38$ | $12.78 \pm 0.30$ | $14.60 \pm 0.27$ | $7.06 \pm 0.40$ | -2.15 |
| 225 | $9.56 \pm 0.18$ | $10.42 \pm 0.28$ | $4.15 \pm 0.33$ | $11.47 \pm 0.85$ | $11.78 \pm 1.07$ | $2.69 \pm 1.07$ | -1.46 |

Ambient and total TI were binned against wind direction (5° bin width) and wake added TI was deduced for each bin. Results are shown on Fig. 10, while numeric values are summarized on Tab. 1. Unfortunately, for the 20% curtailment case, only data on the right side of the wake could be exploited. However, it still provides a very interesting insight of TI reduction with down-regulation as it covers full wake situation (at 210°) to partial wake situations (220 - 225°).

5 It can be observed that the upstream wind turbine curtailment provides a relatively significant decrease in downstream wake added TI, as it is reduced from 14.84% to 12.50% for a wind direction of 210°. As expected, as wind direction increases and we go from full wake situation to partial wake situation, reduction of wake added TI becomes smaller, from 2.34% to 1.46%. In this paper, to model the relationship between percentage reduction of wake added TI with percentage of upstream down-regulation, a linear dependency was assumed for a given wake condition:

10 $$(\%\Delta TI_{wa}) = \frac{\Delta TI_{wa}}{TI_{wa}} = p_{wake}(\%DR). \tag{7}$$

The value of the parameter $p_{wake}$ linking $\Delta TI_{wa}$ with $\%DR$ is expected to be dependent on the relative wind direction between the turbines. Consequently, the parameter for full wake conditions, $p_{fw}$ will differ with the one for partial wake conditions, $p_{pw}$. In this example, where a 20% down-regulation was applied, values of $p_{fw} \approx -0.788$ (from the bin 210°) and $p_{pw} \approx -1.001$ (from the bin 215°) could be computed.

15 In Sect. 5.4.2, dealing with power optimization in a multiple wake case, reduction of wake added TI with upstream curtailment is modeled with the following steps:

1. First, wake added TI at each wind turbine in normal operation case is calculated based on the TI distribution presented in Sect. 5.2 and ambient TI deduced from the most upstream wind turbine.

2. Second, as upstream wind turbines are being gradually curtailed in the optimization process, wake added TI is adjusted based on Eq. 7 above. To simplify the process, only down-regulation of the closest upstream turbine is considered for the

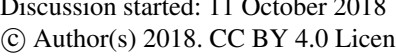



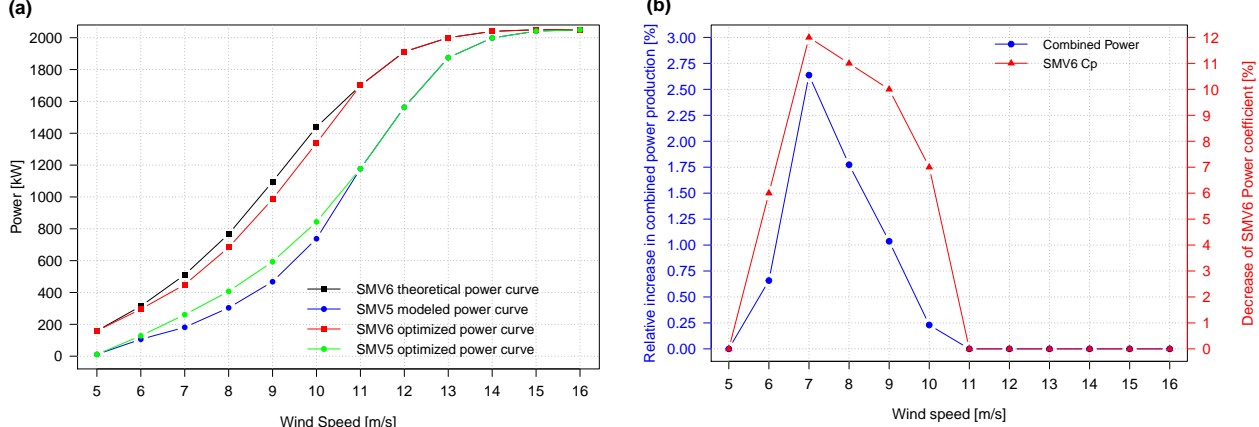

**Figure 11.** Optimization of power production of SMV6 and SMV5. Power curves in base and optimized case (a) and variation of power gain and optimal SMV6 $c_P$ as a function of wind speed for the optimized case (b).

reduction of wake added TI. A similar procedure was followed in Niayifar and Porté-Agel (2016) when modeling wake added TI.

3. Finally total expected TI at the wind turbine is calculated by inverting Eq. 6. This calculated TI can then be given as input into the local TI based calibrated wake model to compute the wake deficit at downstream turbines.

**5.4 Study cases**

**5.4.1 Wind turbines SMV5 and SMV6 (Single Wake case)**

The first case to be studied is the SMV6-SMV5 wake event. It is of particular interest because of the very short spacing between the two wind turbines and their alignment close to prevailing wind directions. For this study case a very simple optimization procedure is used: for each wind speed and relative wind direction, the $c_P$ of SMV6 is gradually decreased up to 20% of its
actual value (and the $c_T$ adjusted consequently based on Eq. 5) and the power production of both wind turbines is computed using the calibrated Jensen model. The optimized power curve for SMV6 is then deduced from all the $c_P$ values giving the best combined production at each wind speed.

Results are presented on Fig. 11 for full wake conditions. In Fig. 11(a) power curves for both wind turbines are plotted while Fig. 11(b) shows the relative increase in combined power production which is obtained at each wind speed with the associated
amount of curtailment which is applied to SMV6. It is observed that the maximum gain represents an increase of about 2.5% and is found at 7 m/s when SMV6 is curtailed by 12% ($c_P$ decreases from 0.46 to 0.405). More generally, it can be seen that the interest of coordinated control is limited to the wind speed range 5 - 11 m/s, i.e. when both $c_P$ and $c_T$ of the upstream turbine are high. In this range, a small decrease in $c_P$ causes a high reduction of $c_T$, as illustrated on the parabolic $c_P$ - $c_T$ relationship

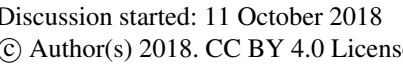
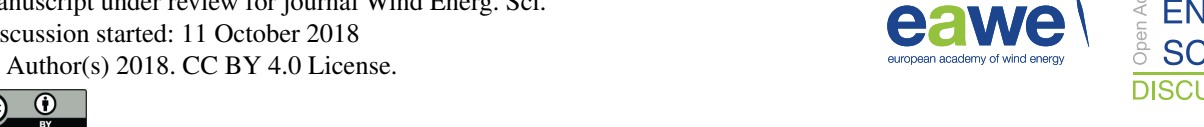

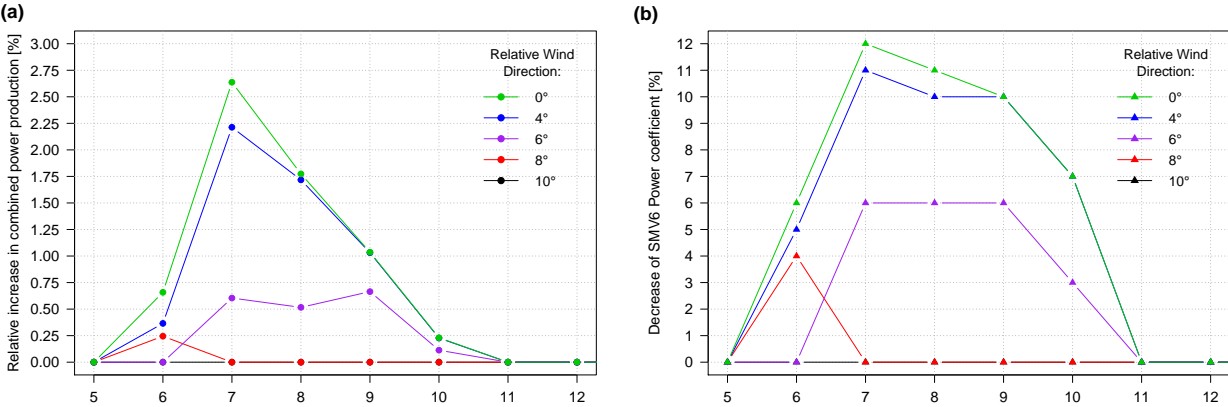

**Figure 12.** Influence of changing relative wind direction. Variation of power gain (a) and optimal SMV6 $c_P$ (b) as a function of wind speed.

of Fig. 8. As wind speed increases further, the upstream wind turbine naturally starts to pitch to limit power production to its nominal value and therefore axial induction control is no longer beneficial.

In Fig. 12 is studied the impact of a changing relative wind direction on the relevance of applying a coordinated control, showing the optimal gain that can be expected (a) and the optimal amount of curtailment required on SMV6 (b) as a function

of the wind speed. It is seen that the wind direction sector on which gains can be observed is particularly narrow. As soon as the full wake condition is no longer respected, i.e. for relative wind direction above $\pm 5°$, the benefit of axial induction control drops almost instantly: when wind direction shifts from 4° to 6° at 7 m/s, gains are reduced from 2.25% to approximately 0.6%. This sensitivity confirms the very limited applicability of a curtailment strategy for power production optimization and the difficulty to implement it in a real case situation. As it is only beneficial on a 10° width wind direction sector centered

around full wake conditions, very stable incoming wind conditions are required to make sure that gains in power production will actually be observed.

### 5.4.2 Row SMV1 to SMV5 (Multiple Wake case)

The second case to be studied is the row SMV1 to SMV5. The power production along the row is now studied for full wake conditions and a wind speed of 8 m/s, for which the coordinated control is expected to give the highest possible gain. Two

different control strategies are being investigated: in the first one only the most upstream turbine (SMV1) is being curtailed while in the second one each wind turbine is optimally curtailed so that total power along the row is maximized. In the first case, the optimum is reached thanks to the same method as in previous section, while for the second case the following procedure (adapted from Heer et al. (2014)) is used:

1. Start with no down-regulation applied on the wind turbines: $\%DR_i = 0\%$ for $i \in \{1 - 5\}$.




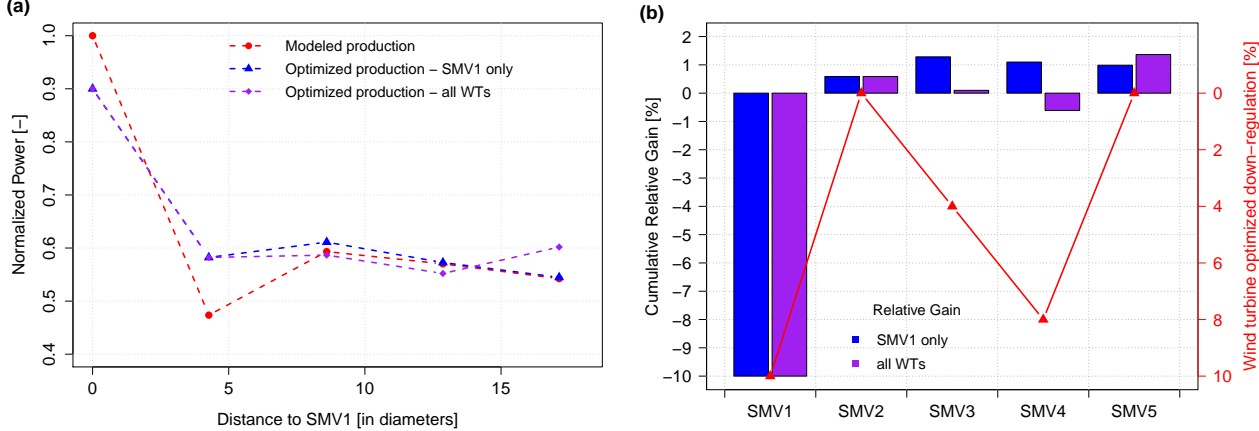

**Figure 13.** Variation of normalized power at each wind turbine (a), cumulative relative gain and optimized down-regulation settings (b) along the row, for full wake conditions (wind direction of 5° at 8 m/s).

2. For wind turbine $i$, from upstream to downstream, find the value $\%DR_i$ maximizing power production along the row and considering that all other $\%DR_{j \neq i}$ remain constant.

3. Repeat step 2 until all $\%DR_i$ stay constant.

Reduction of wake added TI was considered as developed in Sect. 5.3. A value $p_{fw} \approx -0.788$ was used for all wake events
in the row except for SMV2 → SMV3 which corresponds to a partial wake event (alignment of these two turbines is for a wind direction of -2°). A value of $p_{pw} \approx -1.001$ (deduced from wind direction bin 215° in Tab. 1) was used instead.

Figure 13 shows the result of the process, with the normalized power production along the row (a), and the cumulative relative gain and percentage down-regulation at each wind turbine for the second strategy (b) (for the first strategy, same amount of curtailment is applied on SMV1 while all other wind turbines are not curtailed). The cumulative relative gain allows
following the change of power provided by the optimization as we move downstream, it is defined at wind turbine $i$ as:

$$\%RG_i = 100 \frac{\sum_{j=1}^{i} P_j^{opti} - \sum_{j=1}^{i} P_j^{base}}{\sum_{j=1}^{i} P_j^{base}} \tag{8}$$

Consequently the cumulative relative gain at wind turbine SMV5 represents the total gain obtained thanks to the optimization process.

It can be seen from the figures that both strategies lead to an overall increase in power production, with the same amount of
15 curtailment imposed on the most upstream turbine (10% down-regulation). As seen on the cumulative relative gain which is positive at the second turbine, the increase in power at SMV2 is enough to compensate for SMV1 down-regulation. However, for the first strategy, as downstream wind turbines are not curtailed, most of the energy released by SMV1 is captured by SMV2 and SMV3 only. The most downstream turbines SMV4 and SMV5 are only very slightly benefiting from the down-regulation and total relative increase is limited to approximately 1%.





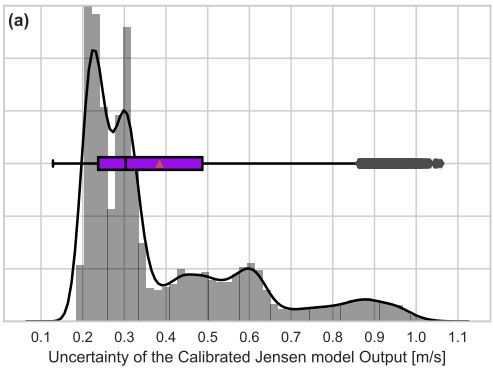
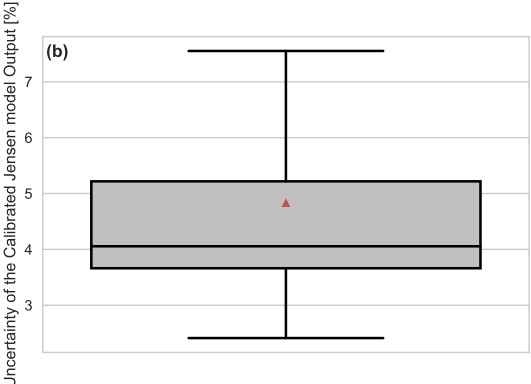

**Figure 14.** Distribution of the uncertainty in the estimated wake velocity by calibrated Jensen model, (a) the histogram and the boxplot of the distribution in units [m/s], (b) the boxplot of the percentage uncertainty distribution. The median, the boxes, the whiskers and the outliers are according to the Tukey's descriptive statistics (Tukey (1962)). The red triangles indicate the mean.

On the contrary, when following the second strategy, the energy made available by SMV1 down-regulation is more equitably shared between all downstream turbines. Indeed, SMV3 and SMV4 are also being curtailed so that high gains can be obtained for SMV5 (as SMV2 is not perfectly aligned with the rest of the row, it is not beneficial to curtail this turbine). As a consequence the cumulative relative gain is decreased at SMV3 and SMV4, but when considering also SMV5 the total relative increase in power production is higher than in the first case, reaching almost 1.5%. This confirms the idea that best gains are obtained when each wind turbine is controlled individually to maximize the overall wind farm output, and not their own power production.

### 5.5 Uncertainty quantification of the calibrated Jensen model

In order to put the optimized wind farm performance into perspective, it is essential to estimate the uncertainty of the model outputs. Since the calibration of the model is based on the operational turbine data, the uncertainty quantification (UQ) is established through the input uncertainty assessment, and its propagation. The uncertainty is defined as the half width of the 68% confidence interval, which corresponds to a distance of a single standard deviation.

In general terms, the uncertainties attached to the SCADA signals are indicated in the IEC standards (IEC et al., 2013), where the main focus is the annual energy production estimates. With the same focus, Gaumond et al. (2012) investigated the wind direction uncertainty in particular, which is the most ambivalent input signal to the modified Jensen model. Within 10-min intervals, the study showed the uncertainty to be at the levels of 5° for Horns Rev-I wind farm. However, since in this study the analysis is based on high frequency data (second-wise data), the documented value of 3° uniform uncertainty in the yaw position signal (IEC et al., 2013) is considered for now. For the wind speed input, the dependency of the uncertainty to the operational range, i.e. region II and III, is shown in Göçmen and Giebel (2018) for effective wind speed, and in IEC et al. (2013) for nacelle wind speed. Here, we consider the conservative estimate of 0.3 m/s with Gaussian distribution. The uncertainty in wind speed is propagated through the estimation of TI (see Sect. 3.3), estimation of $k_w$ (see Eq. 4), estimation




of $c_T$, and finally the estimation of wake deficit through Eq. 1. The resulting uncertainty distributions of the calibrated Jensen model is shown in Fig. 14. The uncertainty is propagated in a continuous time series of 18-hours from Horns Rev-I wind farm, using a Monte Carlo analysis with 1000 realizations per second. The indicators in the boxplots follow Tukey's descriptive statistics (Tukey, 1962) where the boundaries of the whiskers are the lowest and the highest datum within the 1.5 inter-quartile

range, IQR, corresponding to $\pm 2.7\sigma$ from the mean.

Figure 14 shows that the uncertainty of the calibrated model output is not normally distributed. This is mainly due to the fact that along the propagation process, different sources and distributions of uncertainties are convoluted. It is also seen that there are many outliers in the distribution, which occur near the rated wind speed, i.e. around the operational transition between Region II and III. Along that transition region, the assessment of the $c_T$ is highly sensitive to the wind speed (see Fig. 5(b)),

causing the propagation of the uncertainty to diverge. Since for the optimization scenarios discussed above the considered wind speed is lower, the conservative estimate of the model uncertainty can be stated as 0.3 m/s or, more generally, 4%. Compared to the lower gains the developed control strategies show in the previous section, it is very important to note that the modified Jensen model risks to provide the reported power increase in the field tests. In other words, the uncertainty of the model outputs and control inputs needs to be fully analyzed in order to assess the true performance of the wind farm control

approaches. Additionally, in order to see the full benefit of wind farm control schemes, more "accurate" sensors / data, more intelligent methods to analyze the data and different perspectives on how to include the physical complexity of the wind farm flow into the wake models are needed.

Finally, it must also be mentioned that while the axial induction strategy seems to propose a limited benefit in terms of increased combined production compared to other strategies such as wake steering, its impact on wind turbine loads should be

much more profitable. Indeed, thanks to the application of a curtailment on the upstream wind turbine, reduction in thrust and in the tower loads can be expected. Downstream, decrease in the fatigue loading of the turbines can be predicted due to the reduction in the wake added TI, as illustrated on Fig. 10 above. These results are interesting since load reduction can be related to increase lifetime, and therefore decrease in overall cost of energy.

## 6    Conclusions

Field tests are currently being held on a commercial wind farm in France, La Sole du Moulin Vieux, in the scope of a national project. The objectives of these tests are to study the potential of wind farm coordinated control strategy for power optimization and loads reduction. In this paper data from the first campaign was analyzed in detail to propose a modification of the widely used Jensen model. This modification, based on the local measurement of turbulence intensity given by the nacelle anemometer, proved to be enough to improve the accuracy of the model and describe more precisely the individual wake deficit at each wind

turbine. The simplicity of calibration and robustness of the original model is kept since there is still only one parameter to calibrate. This new tuning strategy was validated with data from SMV wind farm but also with data from a row of turbine at Horns Rev-I offshore wind farm.



Using this methodology, power production was optimized in two study cases at SMV wind farm. An axial induction strategy was considered with a model predictive approach, and a $c_T$ estimation procedure was developed in order to assess as accurately as possible the combined evolution of $c_P$ and $c_T$ during down-regulation of the wind turbines. Results from the optimization process show that a gain of 1 to 2% in combined power production can be expected for full wake conditions and wind speed

between 7 - 9 m/s. As wind direction changes or wind speed increases further, gains in power production obtained through the upstream wind turbine down-regulation quickly drop to zero, underlining the limited applicability of an axial induction strategy for power production optimization. Moreover, these gains have to be taken cautiously since some studies have already underlined discrepancies between model predictions and actual power productions of wind turbines, especially when variation of the wake decay constant due to changes in $c_T$ was not taken into account (Annoni et al., 2016). These results are in line

with the performed uncertainty assessment of the modified Jensen model, where the uncertainty is shown to be more than the predicted power increase. This also indicates the importance of an extensive uncertainty quantification on the (simplified, control-oriented) flow models to correctly evaluate the resulting wind farm control strategies.

However, even if the model is not as accurate as it could be, it is hoped that it is still good enough to give indications about optimal settings where gains would possibly be found. Based on the results derived in this paper, a new field campaign was

realized between December 2017 and February 2018 during which a curtailment mode was applied to wind turbine SMV6. Data is currently being processed to determine whether augmentation in combined power production could be achieved. Furthermore data from strain gauges installed in the blades still have to be analyzed to study the impact of axial induction control on wind turbine fatigue loads. Given the quantified uncertainty, even though no gains in power production are obtained, a reduction in loads can be expected.

Future work about wind farm coordinated control realized in the scope of the SMARTEOLE project will also include the analysis of the potential of the wake steering strategy, and the study of the dynamics involved when a wind turbine is curtailed or yawed.

*Competing interests.*  The authors declare that they have no conflict of interest.

*Acknowledgements.*  The authors would like to thank the French national project SMARTEOLE (ANR-14-CE05-0034).




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
