# Peer review of "Local turbulence parameterization improves the Jensen wake model and its implementation for power optimization of an operating wind farm"

_Wind Energy Science, 2018_

## Short Comment (SC1) · 12 Oct 2018

Dear authors,

Thanks for a nice paper, which I read with great interest ! I am listing some comments below, I hope some at least can be useful.

Apologies if the comments are not perfectly neat and well written. Some may just be incorrect or irrelevant, I hope you won't mind if that is the case.

Very interesting paper, much work has been put into this, well done.

All the best, Rémi Gandoin

[Figure]

**1. Introduction**

"in some cases wake effects are still persistent at significant distances downstream": maybe quantify "significant"

"maximize their own power production": this is true only for the constant TSR control region, right ?

"Two different strategies are mainly considered": in this paper ? or in general in the literature ?

"either the upwind turbines are curtailed": does this mean that the ouput power is set to a constant value, or that the power is just decreased compared to the original control settings ?

"small gains in power production are indeed possible": using the first or the second strategy, or both ?

"high variability with incoming wind conditions": - does this mean that they can have positive and negative effect on park production ? or only positive effect but with some (how much ?) variability ? - what is meant by "wind conditions" (speed, direction, temporal/spatial scale) ?

"where wind conditions are fluctuating constantly and significantly": same as above

"Very few full scale field tests have been realized to investigate this question": can you refer to these few, if there are publications available ?

"uncertainties remain high": can you specificy whether you refer to accuracy and/or precision ?

"La Sole du Moulin Vieux (SMV)": 7 x Senvion MM82/2050 ? maybe refer to Figure 1.

"and was dedicated to axial induction control strategy": what does this mean (in a few words) ?

"high level of curtailment": see my question about defining curtailment above.

"could be observed": how ?

"combined power production": you mean the sum of all WTG production ?

"part of the lost power": can this be quantified ?

"the best settings": which parameters are changing ?

"as a function of wind speed and wind direction": measured by the nacelle anemometry ?

"Jensen model": reference ?

"local measurement of turbulence intensity": how ?

"The resulting wake deficit appears to be more consistent with observed data": can you quantify ?

"the original model": you mean the k-parameter value of 0.075 as in http://orbit.dtu.dk/fedora/objects/orbit:66401/datastreams/file_f7da8eb2-e49c-4dc9-9ee5-72846f40ef34/content ?

"Figure 1. Layout of SMV wind farm and location of wind measurement devices. Inter distances between the wind turbines are expressed in rotor diameters": please specify that D=82m. Could we worth showing a bigger area, maybe {49.816789°;2.753916°} to {49.868842°;2.843664°};

"red arrows indicate the main wind direction ": maybe add that these are °N. How wide are the wind-directional bins ? How is the wind direction defined ?

Section 2

"SCADA data": 10-minute ?

Section 3.1

"where U0 is the incoming wind speed at the upstream wind turbine, Uw the velocity in the wake and R the radius of the upstream rotor.2": ,and C_T the Thrust Coefficient corresponding to U0.

"its inaccuracy": which one ?

"local turbulence intensity": see my question above regarding how you measure this.

Section 3.2

General comments: - you may also want to refer to Section 2.4 of (Lissaman, 1976): https://drive.google.com/drive/folders/1tI2p3W1qRj2GsYkt6RI6VITJMpQh7PMc.
- if the wake expansion factor changes within the wind farm, could this be visible on high-resolution wake measurements reported in http://iopscience.iop.org/article/10.1088/1742-6596/1037/7/072008/pdf ? If k=0.9*TI_WTG then, based on Figure 7, the first wake "cone" should expand with an larger angle than the one for the rows downstream, have I understood correctly ?

"This empirical constant is supposed to vary from one wind farm to another": reference ?

"or vice versa": why is that ?

Section 3.3

"Four months of second-wise SCADA data": you mean 1-Hz ?

"Figure 3": - could you use a function using "density scatter plots" in Matlab and plot the mean and median binned values as well? - the plot NWS vs Met mast shows less scatter than the plot NWS vs LiDAR: could you also show Met mast vs LiDAR (there maybe 10-minute time offset) ?

Section 4.1

Ggeneral comments: - have you considered using the M2 mast dataset measurements for Horns Rev 1? - have you considered a wider wind directional bin ? As I remember, Gaumond showed that using narrow bins, led to bias in the validation, since the model is "steady state" and will consider the turbines always aligned. A workaround is to run a model simulation every $0.5°$ and then weight the results using a gaussian distribution of the wind directions.

"modified Jensen model": maybe only semantics here, but it seems to me that you are only tuning the input k parameter and not changing the model.

"Region II": can you highlight this region in Figure 5 and Figure 2 ?

"Figure 6": - could you state that "TI" is the TI measured using the Nacelle Anemometer (and not the ambient TI) ? - could you add a curve which uses k_w=0.4TI_ambient, as suggested in the Sexbierum paper ? - the TI value of 12 % for the first WTG at HR1 seem large, for offshore conditions - see typical values in https://pcwg.org/proceedings/2014-10-06/06-Turbulence-Intensity-measmnts-offshore-4-PC-verification-wind-res-assmt-R-RiveraLamatA-D-Pollack-Dong.pptx - can you show a histogram of the corresponding wind speeds ? - for (b): do you see a difference between nightime and daytime ?

"was clearly overestimating the power deficit for the wind farm of Horns Rev-I, as it 10 gives narrower wake growth within the wind farm": it depends actually, see http://www.eera-dtoc.eu/wp-content/uploads/files/Nygaard_Systematic_quantification_of_wake_model_uncertainty_offshore2015presentation.pdf. The stability conditions may well differ significantly between the different studies, for the same wind farm, since people use different datasets for model validation.

"Figure 7": - the value of TI for the first WTG at HR1 is 8 %, is was 12% in Figure 6 - can you explain why this is ? - same as for Figure 6: could you state that "TI" is the TI measured using the Nacelle Anemometer (and not the ambient TI) ? - same as for Figure 6: could you add a curve which uses k_w=0.4TI_ambient, as suggested in the

Sexbierum paper ?

Section 5.1

About eq. (5): once the WTG is curtailed, will this relationship hold ? See for instance Figure 1 of http://iopscience.iop.org/article/10.1088/1742-6596/1037/3/032039/pdf (I don't have the answer).

Section 5.2

"The reason for this increase was related the presence of the motorway": could it be the large warehouse located at $330°$, 2km upstream of the mast ? Could you make a plot on nighttime and on daytime ?

Section 5.3

"as the upstream wind turbine is down-regulated, the wake added TI emitted by this turbine is reduced":  because it is a function of CT, and CT is reduced ?   The TI_added is also a function of the downstream distance, see chap 3 of http://orbit.dtu.dk/fedora/objects/orbit:79899/datastreams/file_269c3f19-0001-4e41-b754-b5b322a826cb/content.

Table 1: could you also show the relative power values, for a given wind speed bin (for instance 7 m/s) ?

Section 5.4.1

"It is observed that the maximum gain represents an increase of about 2.5% and is found at 7 m/s when SMV6 is curtailed by 12% (cP decreases from 0.46 to 0.405)":as I understand, in this situation, you reduce C_p by x%, and derive a new C_t using eq. (5). Then, you choose a value of k using eq. (7). Can you then also plot these new and old values of C_t and k, that are used the calculation, in Figure 12 ?  Or provide a worked-out example ? It may help the reader understand what goes on in the calculation.

"very stable incoming wind conditions": you mean stable atmosphere ?

---

## Short Comment (SC2) · 12 Oct 2018

There was at least one typo in my comments - apologies:

Section 3.2: read "If k=0.9*TI_WTG then, based on Figure 7, the first wake "cone" should expand with a **smaller** angle than the one for the rows downstream, **since the TI of the first WTG is smaller**"

Rémi
* * *

---

## Referee Comment (RC1) · Anonymous Referee #1 · 6 Nov 2018

This manuscript deals with an ad-hoc tuning of the expansion parameter of the Jensen model as a function of the incoming turbulence intensity. The calibrated model is then used for the power optimization of a wind farm through a coordinated axial-induction control. The concept is not new, but the application to a real wind farm is very interesting. I would add some comments specifying that the variability in wind direction, thus of wake interactions, and the daily cycle of the atmospheric stability, might lead to a more complicated tuning of the model for real applications, or at least for a real implementation. I found a recent paper with a very similar approach to this, Santhanagopalan et al. 2018 Renewable Energy, 116, 232-243. In that paper, the authors used a RANS model to perform axial-induction optimization of a turbine array for the different incoming wind

turbulence. They used as objective function the maximization of the power capture with a penalization due to the fatigue loads derived by the wake-generated turbulence. It will be interesting to see how these results compare to those presented in this manuscript.

---

## Referee Comment (RC2) · Anonymous Referee #2 · 24 Nov 2018

This paper proposes a method for tuning the Jensen wake model for a wind turbine based upon the locally measured turbulence intensity. The work is based upon field test data at the wind farm La Sole du Moulin Vieux (SMV) in France. The tuned Jensen wake model is then used to optimize the power settings of the wind turbines to maximize the wind farm power capture; this optimization is similar to other wind farm optimizations that have been reported in the literature. The study in this paper considers a single wake case where there is only one wind turbine in the wake of another turbine, and the study also considers a multiple wake case where there is a row of 5 wind turbines where each turbine subsequently wakes (either fully or partially) the following turbine. Finally, the uncertainty of the calibrated Jensen model is quantified, showing that more work remains to be done before the tuned Jensen model can provide guarantees of increases in wind farm power production.

Overall, the paper is reasonably written and there may be useful results to be published, though the conclusiveness of the proposed wind farm control approach is still unclear due to the uncertainty that remains in the tuned Jensen model. As such, I would recommend that a much shortened paper summarizing the tuned Jensen model results based on field data and the uncertainty quantification, and indicating that more studies are needed before the tuned Jensen model can be reliably used for coordinated wind farm control based upon axial induction control strategies. The challenges of axial induction control strategies have already been reported in (Annoni et al., 2016), among other papers.

Some more specific comments and suggestions:

1. Yellow is used for one of the curves in several of the figures. Because the yellow curves are very difficult to see, I would highly recommend replacing each yellow curve with another choice of colored curve, perhaps with a different shaped marker on the curve to distinguish it from other curves in the same figure.

2. In Figure 9, the curves for both SMV1 and SMV7 are nearly identical in color. The curves for SMV3 and SMV4 are also very similar in color. I would recommend at least choosing different shaped markers (not all circles) to help distinguish these curves in the Figure.

3. There are small grammatical mistakes or typos throughout the manuscript, and it is recommended that the authors more carefully proofread subsequent submissions.

---

## Author Comment (AC1) · 7 Jan 2019

**Reviewer comments**

Thomas Duc

January 7, 2019

**1 Response to short comments 1 and 2**

Dear authors,

Thanks for a nice paper, which I read with great interest ! I am listing some comments below, I hope some at least can be useful. Apologies if the comments are not perfectly neat and well written. Some may just be incorrect or irrelevant, I hope you won't mind if that is the case. Very interesting paper, much work has been put into this, well done.

All the best, Rémi Gandoin

*Dear Rémi Gandoin, thank you very much for taking the time to review the article. We appreciate your input and your interest in our work! You can find detailed responses to your comments down below.*

1. Introduction

"in some cases wake effects are still persistent at significant distances downstream": maybe quantify "significant"

*A quantification of 10 to 15 D, which is taken from the publication of Sanderse, is now provided.*

"maximize their own power production": this is true only for the constant TSR control region, right ?

*Not exactly: this is true that the efficiency of a wind turbine is maximum in the region 2 of the power curve, but this is not the point of the sentence. This sentence essentially means that with current state-of-the-art control, for a given wind velocity, a wind turbine will try to extract as much power as it can from the wind by applying its power curve and will not consider what is happening downstream. In contrast, with coordinated control, the power of upstream wind turbine is purposely de-rated to increase the power of downstream turbines.*

"Two different strategies are mainly considered": in this paper ? or in general in the literature ?

*It is meant in general in the literature, e.g. see the paper from Knudsen. Some other strategies have also been studied, like tilt control or individual pitch control but they seem currently less feasible and much less popular in wind farm control research. In this paper, the focus is only on axial induction control.*

"either the upwind turbines are curtailed": does this mean that the output power is set to a constant value, or that the power is just decreased compared to the original control settings ?

*This means that output power is just decreased compared to the original control settings.*

"small gains in power production are indeed possible": using the first or the second strategy, or both ?

*Apparently from both strategies, but some latest results tends to indicate that the yaw offset (or wake steering) strategy have much better potential for increasing power production.*

"high variability with incoming wind conditions": - does this mean that they can have positive and negative effect on park production ? or only positive effect but with some (how much ?) variability ? - what is meant by "wind conditions" (speed, direction, temporal/spatial scale) ?

"where wind conditions are fluctuating constantly and significantly": same as above

*Depending on the wind conditions, they might have a positive or a negative impact on power production. The most critical parameter is wind direction but as underlined in this paper wind speed and turbulence intensity are also very important. See e.g. later on section 5 : for axial induction control, there is a very limited range of wind speed and direction for which the strategy is profitable. If the wind direction changes but the turbine does not react quickly enough and curtailment is still applied on the wind turbine, then there will be loss in aggregated power production because the upstream curtailment will not create any (or sufficient) gain downstream.*

"Very few full scale field tests have been realized to investigate this question": can you refer to these few, if there are publications available ?
*These references are already listed in the following sentence of the paper.*

"uncertainties remain high": can you specify whether you refer to accuracy and/or precision ?
*The uncertainty here is related to both the accuracy and adequacy of the model (i.e. simplified steady-state model to be evaluated against dynamic conditions in the field) as well as the precision of the measurements (i.e. how 'trustworthy' the SCADA readings are and how the measurement period affects the overall assessment of the model performance). It is shown in the uncertainty quantification section that the gains expected are small with respect to the scatter of experimental data points. So it is required to have longer periods of measurements with higher precision where possible, to make sure that there is a positive impact in the production.*

"La Sole du Moulin Vieux (SMV)": 7 x Senvion MM82/2050 ? maybe refer to Figure 1.
*The wind farm is described in detail in section 2, we added a mention to this when presenting the outline of the paper at the end of the introduction.*

"and was dedicated to axial induction control strategy": what does this mean (in a few words) ?
*The upstream wind turbines are curtailed so that their axial induction is decreased. This leaves a more energetic flow downstream that can be beneficial for the turbines in the wake.*

"high level of curtailment": see my question about defining curtailment above.
*The upstream wind turbine was de-rated by 20%, see section 5.3.*

"could be observed": how ?
*This was done by the analysis of downstream active power data at the downstream turbine. This is now mentioned in the paper.*

"combined power production": you mean the sum of all WTG production ?
*Yes, in this case it there were only two wind turbines. We changed the "combined" into "aggregated" to make the sentence clearer.*

"part of the lost power": can this be quantified ?
*This is actually a bit difficult to quantify because it really depends on the wind speed and wind direction bins. In the revised version of the paper I added as reference my master thesis, in which more details can be found in section 6.1. if you are interested.*

"the best settings": which parameters are changing ?
*This was a bit misleading as in practice we do not have access to control parameters. This has been changed into "the optimal de-rating to be applied as a function...".*

"as a function of wind speed and wind direction": measured by the nacelle anemometry ?
*In general, any sensor able to measure the incoming wind speed and direction and send this information to the wind turbine controller could be used, but in our field tests only nacelle anemometry*

*is involved for the optimized control.*

"Jensen model": reference ?
*The reference to the Jensen model is now added in the introduction.*

"local measurement of turbulence intensity": how ?
*This is provided by the nacelle anemometer. This is now indicated on the revised version of the paper.*

"The resulting wake deficit appears to be more consistent with observed data": can you quantify ?
*Some figures are already provided in the abstract and will be discussed in more details in section 4.*

"the original model": you mean the k-parameter value of 0.075 as in `http://orbit.dtu.dk/fedora/objects/orbit:66401/datastreams/file_f7da8eb2-e49c-4dc9-9ee5-72846f40ef34/content` ?
*Yes, in the revised version it is now written "when using a constant value" to clarify this.*

"Figure 1. Layout of SMV wind farm and location of wind measurement devices. Inter distances between the wind turbines are expressed in rotor diameters": please specify that D=82m.
*The mention of the diameter is now added in the revised version.*

Could we worth showing a bigger area, maybe {49.816789°;2.753916°} to {49.868842°;2.843664°};
*As it would further elongate the paper, as was criticized by the other reviewers, the broader location map is skipped for now. Additionally, the added value of such figure would be limited and the readers could always refer to other sources (e.g. Google Earth) to see the details of the area of the wind farm if interested.*

"red arrows indicate the main wind direction ": maybe add that these are °N. How wide are the wind-directional bins ? How is the wind direction defined ?
*Reference for 0° in wind energy is conventionally absolute north, so would be redundant to mention in the article. Regarding bin width, they are defined later in section 4.*

Section 2
"SCADA data": 10-minute ?
*Both 10-minutes and 1-second data are studied, see later in the paper.*

Section 3.1
"where U0 is the incoming wind speed at the upstream wind turbine, Uw the velocity in the wake and R the radius of the upstream rotor.2": ,and $C_T$ the Thrust Coefficient corresponding to U0.
*$c_T$ is already defined above the equation.*

"its inaccuracy": which one ?
*The inaccuracy of the Jensen model.*

"local turbulence intensity": see my question above regarding how you measure this.
*This is now corrected following the comment above.*

Section 3.2
General comments: - you may also want to refer to Section 2.4 of (Lissaman, 1976): `https://drive.google.com/drive/folders/1tI2p3W1qRj2GsYkt6RI6VITJMpQh7PMc`.
*This reference has been added to the revised version.*

- if the wake expansion factor changes within the wind farm, could this be visible on high-resolution wake measurements reported in `http://iopscience.iop.org/article/10.1088/1742-6596/1037/7/072008/pdf` ?

*It would be interesting to look at as a separate study, although it might be a bit difficult to see the change in wake expansion because of the superposition of the wakes emitted by the different upstream turbines.*

If $k = 0.9 * TI_{WTG}$ then, based on Figure 7, the first wake "cone" should expand with an smaller angle than the one for the rows downstream, since the TI of the first WTG is smaller, have I understood correctly ?
*Yes, exactly.*

"This empirical constant is supposed to vary from one wind farm to another": reference ?
*Well, this is actually seen later in the paper: we have a value of 0.075 for SMV wind farm, but 0.09 for Horns Rev (while generally values of 0.04 and 0.05 are used for offshore wind farms).*

"or vice versa": why is that ?
*This depends on how the model has been calibrated and the particular layout of the wind farm.*

Section 3.3
"Four months of second-wise SCADA data": you mean 1-Hz ?
*Yes, this has been clarified in the revised version.*

"Figure 3": - could you use a function using "density scatter plots" in Matlab and plot the mean and median binned values as well?
*In this graph we prefer to stay with the scatter plot in order to be consistent with the reference Göçmen and Giebel (2016) that we are using in the discussion.*

- the plot NWS vs Met mast shows less scatter than the plot NWS vs LiDAR: could you also show Met mast vs LiDAR (there maybe 10-minute time offset) ?
*The plot Met-mast vs Lidar is already there : it is the top left graph. The scatter is explained by the fact that sensors are not facing the same wake effects since measurement do not occur at the same location. The Met-mast is less affected than the windcube to wake effects because it is located further away from the farm.*

Section 4.1
General comments: - have you considered using the M2 mast dataset measurements for Horns Rev 1?
*First and foremost, during the period where the 1Hz SCADA from Horns Rev-I was available, M2 was already out of operation. Additionally, the idea was to focus only on SCADA data in order to make this study easily reproducible for any wind farms, as especially offshore a met mast is often not feasible/available. To be consistent, the met-mast data at SMV wind farm is only used to validate the TI measurement from SCADA, where the rest of the analysis is based on.*

- have you considered a wider wind directional bin ? As I remember, Gaumond showed that using narrow bins, led to bias in the validation, since the model is "steady state" and will consider the turbines always aligned. A workaround is to run a model simulation every 0.5°and then weight the results using a gaussian distribution of the wind directions.
*We considered using a larger bin width but finally decided to stay with that one to ensure a consistency with the results found later in the paper. Indeed in section 5.4.1., it is shown that axial induction control is only beneficial on a $\pm 5°$wind direction sector centered on full wake conditions. The objective in this section was essentially to compare the two calculation procedures in the same conditions, i.e. with the same bin width and no gaussian averaging. It is expected that any post processing that will improve the behavior of the original calculation procedure will also benefit to the new calculation procedure.*

"modified Jensen model": maybe only semantics here, but it seems to me that you are only tuning the input k parameter and not changing the model.

*Yes this is right. In the revised version we changed "modified" into "tuned".*

"Region II": can you highlight this region in Figure 5 and Figure 2 ?
*These regions are defined two paragraphs later, adding them also here might just overload the figure.*

"Figure 6": - could you state that "TI" is the TI measured using the Nacelle Anemometer (and not the ambient TI) ?
*This has been added in the figure caption.*

- could you add a curve which uses $k_w = 0.4 TI_{ambient}$, as suggested in the Sexbierum paper ?
*We tried to add this particular curve but it did not fit well with the experimental data we had (the modeled wake deficit was too high). Thus we decided to remove it in order not to overload the figure with a curve that would not be used in the discussion. For your curiosity, I am just adding in this document the figure with this particular curve.*

[Figure]

Figure 1: Comparison of the model performance at SMV wind farm. Variation of experimental and modeled normalized power and turbulence intensity (measured by the nacelle anemometer) along the row. Error bars indicates the 68% normalized confidence interval.

- the TI value of 12% for the first WTG at HR1 seem large, for offshore conditions - see typical values in `https://pcwg.org/proceedings/2014-10-06/06-Turbulence-Intensity-measmntsoffshore-4-PC-verification-wind-res-assmt.pptx` - can you show a histogram of the corresponding wind speeds ?
*There is a misunderstanding here: Figure 6 is about SMV onshore wind farm, so a value of 12% is very consistent.*

- for (b): do you see a difference between nightime and daytime ?
*We did not investigate this, it goes a bit beyond the scope of this paper (see the new added section 5.6) but it might be an interesting point to study in the future.*

"was clearly overestimating the power deficit for the wind farm of Horns Rev-I, as it 10 gives narrower wake growth within the wind farm": it depends actually, see `http://www.eera-dtoc.eu/wpcontent/uploads/files/Nygaard_Systematic_quantification_of_wake_model_uncertainty_offshore2015presentation.pdf`. The stability conditions may well differ significantly between the different studies, for the same wind farm, since people use different datasets for model validation.
*In the reference of Nygaard et al, the wind farms are anonymous so it is not possible to conclude anything about Horns Rev-I wind farm particularly. Still, the changes in stability even for the same*

*wind farm is simply a physical phenomena which shows again the importance of uncertainty quantification as the model performance assessment depends on the period of the input and validation dataset. A way to reduce those uncertainties might be to "dynamically" calibrate the Jensen model. One can find different wake expansion coefficients for different seasons, etc. However interesting the concept is, those analyses are beyond the scope of this paper.*

"Figure 7": - the value of TI for the first WTG at HR1 is 8 %, is was 12% in Figure 6 - can you explain why this is ? - same as for Figure 6: could you state that "TI" is the TI measured using the Nacelle Anemometer (and not the ambient TI) ?
**Both these comments are adressed above.**

- same as for Figure 6: could you add a curve which uses $k_w = 0.4TI_{ambient}$, as suggested in the Sexbierum paper ?
**Same as above, the modeled wake deficit using $k_w = 0.4TI_{ambient}$ did not really fit the experimental data we had. Here is the corresponding figure.**

[Figure]

Figure 2: Comparison of the model performance at Horns Rev-I wind farm (row 5). Variation of experimental and modeled normalized power and turbulence intensity (measured by the nacelle anemometer) along the row. Error bars indicates the 68% normalized confidence interval.

Section 5.1
About eq. (5): once the WTG is curtailed, will this relationship hold ? See for instance Figure 1 of `http://iopscience.iop.org/article/10.1088/1742-6596/1037/3/032039/pdf` (I don't have the answer).
**This equation is derived from analysis of guaranteed curtailment modes so it is expected (to our best knowledge) that it will hold.**

Section 5.2
"The reason for this increase was related the presence of the motorway": could it be the large warehouse located at 330°, 2km upstream of the mast ?
**It could be, but the motorway is closer, and seems also to explain the increase in SMV1 TI observed at 5°.**

Could you make a plot on nighttime and on daytime ?
**As already mentioned above, we did not investigate difference between day and night. In Figure 9 the point is essentially to show that nacelle anemometer TI is capturing wake effects for all wind turbines and outside of any wake effects it is consistent with ambient TI measured by the mast.**

Section 5.3

"as the upstream wind turbine is down-regulated, the wake added TI emitted by this turbine is reduced": because it is a function of CT, and CT is reduced ?

*Yes, exactly.*

The $TI_{added}$ is also a function of the downstream distance, see chap 3 of `http://orbit.dtu.dk/fedora/objects/orbit:79899/datastreams/file_269c3f19-0001-4e41-b754-b5b322a826cb/content`.

*Yes, but in this case downstream distance remains constant as we are studying two wind turbines.*

Table 1: could you also show the relative power values, for a given wind speed bin (for instance 7 m/s) ?

*Here there are no wind speed bins involved but only a binning against wind direction.*

Section 5.4.1

"It is observed that the maximum gain represents an increase of about 2.5% and is found at 7 m/s when SMV6 is curtailed by 12% (cP decreases from 0.46 to 0.405)":as I understand, in this situation, you reduce $C_p$ by x%, and derive a new $C_t$ using eq. (5). Then, you choose a value of k using eq. (7). Can you then also plot these new and old values of $C_t$ and k, that are used the calculation, in Figure 12 ? Or provide a worked-out example ? It may help the reader understand what goes on in the calculation.

*Yes these are the steps we are following. In order not to surcharge the plot, we added a small sentence telling how much $c_T$ is reduced when $c_P$ is decreased. We hope this will improve the clarity of the paper.*

"very stable incoming wind conditions": you mean stable atmosphere ?

*No I mean stable wind direction and wind speed, in order to stay in the window for which coordinated control is profitable. If wind direction is changing too frequently, the turbine might not react quick enough to these changes to apply the optimal control at each instant. To clarify, this point, we changed the word "stable" into "steady".*

**2 Response to anonymous referee 1**

This manuscript deals with an ad-hoc tuning of the expansion parameter of the Jensen model as a function of the incoming turbulence intensity. The calibrated model is then used for the power optimization of a wind farm through a coordinated axial-induction control. The concept is not new, but the application to a real wind farm is very interesting.

I would add some comments specifying that the variability in wind direction, thus of wake interactions, and the daily cycle of the atmospheric stability, might lead to a more complicated tuning of the model for real applications, or at least for a real implementation.

*First of all, thank you for your contributions and comments regarding the review of our work. Indeed, this is an important point. We reorganized a bit the paper to add a new section 5.6 in which we discuss shortly these issues related to practical implementation of the axial induction strategy in the field.*

I found a recent paper with a very similar approach to this, Santhanagopalan et al. 2018 Renewable Energy, 116, 232-243. In that paper, the authors used a RANS model to perform axial-induction optimization of a turbine array for the different incoming wind turbulence. They used as objective function the maximization of the power capture with a penalization due to the fatigue loads derived by the wake-generated turbulence. It will be interesting to see how these results compare to those presented in this manuscript.

*Thank you for indicating us this paper, which is very related to our study. We thus added this reference in the introduction and in the discussion about the tuning of the Jensen model. The in-depth comparison between the results from the two papers might indeed be an interesting study to realize in the future.*

**3 Response to anonymous referee 2**

This paper proposes a method for tuning the Jensen wake model for a wind turbine based upon the locally measured turbulence intensity. The work is based upon field test data at the wind farm La Sole du Moulin Vieux (SMV) in France. The tuned Jensen wake model is then used to optimize the power settings of the wind turbines to maximize the wind farm power capture; this optimization is similar to other wind farm optimizations that have been reported in the literature. The study in this paper considers a single wake case where there is only one wind turbine in the wake of another turbine, and the study also considers a multiple wake case where there is a row of 5 wind turbines where each turbine subsequently wakes (either fully or partially) the following turbine. Finally, the uncertainty of the calibrated Jensen model is quantified, showing that more work remains to be done before the tuned Jensen model can provide guarantees of increases in wind farm power production.

Overall, the paper is reasonably written and there may be useful results to be published, though the conclusiveness of the proposed wind farm control approach is still unclear due to the uncertainty that remains in the tuned Jensen model. As such, I would recommend that a much shortened paper summarizing the tuned Jensen model results based on field data and the uncertainty quantification, and indicating that more studies are needed before the tuned Jensen model can be reliably used for coordinated wind farm control based upon axial induction control strategies. The challenges of axial induction control strategies have already been reported in (Annoni et al., 2016), among other papers.

*Thank you very much for your time and effort in reviewing this paper - we believe that the clarity is improved in the revised version. We agree that axial induction control has been a fairly widely studied topic over the past years, but most of the work was focused on high fidelity simulations. To our knowledge, it has very rarely been applied to a real and operating wind farm using a wake model calibrated with actual SCADA data in the scope of the realization of field tests. The principle of this paper is also to propose methodologies that will help for the implementation of wind farm control in the field. Furthermore, we also think that the $c_P$ / $c_T$ estimation procedure and the uncertainty quantification of the model are also nice results to show and they are better integrated with the rest of the paper if they can be illustrated and put in perspective with optimization study cases in the farm.*
*Consequently we think that the section 5 adds value and perspective to the paper about wind farm optimization. In order to address your remarks and reservations, we added a section 5.6 in which we discussed in more detail the limitations of the model and the requirements for a practical implementation of axial induction control in the field.*

Some more specific comments and suggestions:
1. Yellow is used for one of the curves in several of the figures. Because the yellow curves are very difficult to see, I would highly recommend replacing each yellow curve with another choice of colored curve, perhaps with a different shaped marker on the curve to distinguish it from other curves in the same figure.

*Thank you for pointing out this issue, we updated the graphs to take into account you recommendations.*

2. In Figure 9, the curves for both SMV1 and SMV7 are nearly identical in color. The curves for SMV3 and SMV4 are also very similar in color. I would recommend at least choosing different shaped markers (not all circles) to help distinguish these curves in the Figure.

*Same as above, Figure 9 has been modified to change the colors of the indicated curves and set a different marker for all curves.*

3. There are small grammatical mistakes or typos throughout the manuscript, and it is recommended that the authors more carefully proofread subsequent submissions.

*We read again the manuscript and tried to correct all these small mistakes.*

[revised manuscript text omitted]

---

## Author Response (AR2)

**Reviewer comments**

Thomas Duc

March 11, 2019

**1  Minor revision suggestions from anonymous referee 2**

The authors have definitely improved the clarity of the paper, and I agree that the actual wind farm experimental data make this a worthwhile paper.

*Thank you very much for for this second review of our paper, and your suggestions to further improve its clarity. Please see below the answers to your questions and recommendations.*

I still have several suggestions for further improving the paper:

1. The authors should reference Figure 6 in the last paragraph on Page 9.

2. In Figure 6(a), since there are left (power) and right (TI) axes, I would suggest augmenting the two legend labels with "Jensen, $k_w = \ldots$" to be "Estimated Power using Jensen, $k_w = \ldots$" since otherwise readers may not be sure if those curves are to be read with the left or right axis.

In Figure 6(b), it may be beneficial for readers if the curves for $k_w = 0.75*\text{TI}$ and $k_w = 0.075$ have the same color, line, and marker types as the corresponding curves in Figure 6(a).

3. Similar comments apply to Figure 7: Augment the two legend labels with "Jensen, $k_w = \ldots$" to be "Estimated Power using Jensen, $k_w = \ldots$"

In Figure 7(b), it would helpful if the curves for $k_w = 0.9*\text{TI}$ and $k_w = 0.09$ have the same color, line, and marker types as the corresponding curves in Figure 7(a).

4. Figure 8: I would suggest using "Above rated WS" rather than "After rated WS", and similarly in the text discussion.

*For all the listed points above 1 - 4, we modified the figures and the text of the paper to take into account your recommendations.*

5. Equation (7): Is the denominator $TI_{wa}$ for the case when there is no curtailment? The right-most column in the case with curtailment in Table 1 is also labeled $TI_{wa}$.

*Thank you for this remark, indeed this point was not clear before. We now labelled the right-most column of Table 1 $TI_{wa}^{curt}$ to distinguish the two cases "curtailment" and "no curtailment". Consequently $TI_{wa}$ in Equation (7) represents the case with no curtailment, since this relationship aims to describe the reduction of wake added TI allowed by the turbine curtailment with respect to the case when no curtailment is applied on the turbine.*

6. Section 5.5: When you use the word "conservative" in "conservative estimate of 0.3 m/s", do you mean that 0.3 m/s is a higher wind speed error than would be expected? Or a lower wind speed error than would be expected?

*The information has now been clarified in the manuscript by Here, we consider the conservative estimate of 0.3 m/s with Gaussian distribution, where the expected error is less with 90% likelihood below rated region.*

7. Page 20, last paragraph of Section 5.5: "Figure 14 shows that the uncertainty of the calibrated model output is not normally distributed. ... It is also seen that there are many outliers in the distribution, which occur near the rated wind speed, i.e. around the operational transition between Region II and III." How is it possible to know

or "see" that the outliers in Figure 14 occur near rated wind speed? There are no absolute wind speeds labeled in Figure 14.

**You are right, this cannot be seen in Figure 14. This study was in fact done in the previously cited paper, Göçmen and Giebel (2018). We added this reference in the sentence to clarify this point.**

8. I would suggest removing ellipses (three dots) in a formal journal article:

a. On page 21, line 2.

b. On Page 23, line 26. I'm actually not sure why " . . . " appears twice in this reference.

*The ellipses have been removed. Regarding the second point, this was due to a small problem in the bibtex reference that has now been fixed. Thank you!*

[revised manuscript text omitted]